psychology/behaviour

COVID-19, misinformation, fake news, vaccine hesitancy

**Author for correspondence:**
Sander van der Linden
e-mail: sander.vanderlinden@psychol.cam.ac.uk

# Susceptibility to misinformation about COVID-19 around the world

Jon Roozenbeek[1,2], Claudia R. Schneider[1,3], Sarah Dryhurst[3], John Kerr[1,3], Alexandra L. J. Freeman[3], Gabriel Recchia[3], Anne Marthe van der Bles[1,3,4] and Sander van der Linden[1,3]

[1]Department of Psychology, University of Cambridge, Downing St., CB2 3EB Cambridge, UK
[2]Section of Slavonic Studies, University of Cambridge, Sidgwick Avenue, CB3 9DA Cambridge, UK
[3]Winton Centre for Risk and Evidence Communication, University of Cambridge, Wilberforce Road, CB3 0WA Cambridge, UK
[4]Department of Psychology, University of Groningen, Grote Kruisstraat 2/1, 9712 TS Groningen, The Netherlands

  JR, 0000-0002-8150-9305; CRS, 0000-0002-6612-5186; SD, 0000-0002-7772-8492; JK, 0000-0002-6606-5507; ALJF, 0000-0002-4115-161X; GR, 0000-0002-0210-8635

Misinformation about COVID-19 is a major threat to public health. Using five national samples from the UK ($n = 1050$ and $n = 1150$), Ireland ($n = 700$), the USA ($n = 700$), Spain ($n = 700$) and Mexico ($n = 700$), we examine predictors of belief in the most common statements about the virus that contain misinformation. We also investigate the prevalence of belief in COVID-19 misinformation across different countries and the role of belief in such misinformation in predicting relevant health behaviours. We find that while public belief in misinformation about COVID-19 is not particularly common, a substantial proportion views this type of misinformation as highly reliable in each country surveyed. In addition, a small group of participants find common factual information about the virus highly unreliable. We also find that increased susceptibility to misinformation negatively affects people's self-reported compliance with public health guidance about COVID-19, as well as people's willingness to get vaccinated against the virus and to recommend the vaccine to vulnerable friends and family. Across all countries surveyed, we find that higher trust in scientists and having higher numeracy skills were associated with lower susceptibility to coronavirus-related misinformation. Taken together, these results demonstrate a clear link between susceptibility to misinformation and both vaccine hesitancy and a reduced likelihood to comply with health guidance measures, and suggest that interventions which aim to improve critical thinking and trust in science may be a promising avenue for future research.

# 1. Introduction

The first human infection with the SARS CoV-2 novel coronavirus (COVID-19) was reported in December 2019 in Wuhan, China. Within three months, the virus spread across the globe igniting a global public health emergency. As of September 2020, there have been over 25 million cases worldwide and more than 800 000 people have died from the virus [1]. Scholars increasingly recognize that in order to contain the spread of the virus, insights from the social and behavioural sciences play an important role, especially when it comes to the spread of misinformation about the virus [2]. Indeed, misinformation about the COVID-19 pandemic is a serious threat to both public health and international relations, ranging from the proliferation of damaging health advice, such as ingesting bleach, to politically motivated conspiracies about where the virus originated from. In fact, the proliferation of false and misleading information about the virus, how it spreads, how to cure it and who is 'behind' it, has prompted the World Health Organization to warn of an ongoing 'infodemic' [3,4]. For example, false conspiracy theories about 5G masts causing or exacerbating COVID-19 symptoms have led to people setting mobile phone masts ablaze, endangering lives and property [5,6].

Prior research on the spread of misinformation in general has mostly been confined to the context of the 2016 US presidential election, finding that exposure to fake news was fairly limited in the population [7–10]. Similarly, large-scale analyses of Facebook data in the context of vaccine hesitancy also find that anti-vaccination groups are currently in the minority. However, the projected growth in anti-vaccination views is expected to dominate online discourse within a decade without intervention [11]. In the context of COVID-19, a recent analysis of the most viewed coronavirus YouTube videos found that over 25% of the top videos contained misleading information and totalled 62 million views worldwide [12]. There is also evidence to suggest that exposure to misinformation about the virus may be more common than often assumed. For example, a poll by Ofcom in the UK found that almost half (46%) of the British population report exposure to fake news about the coronavirus [13]. In particular, among those exposed, nearly two-thirds (66%) report seeing it on a daily basis, which is problematic as repeated exposure is known to increase belief in fake news [14].

The proliferation of misinformation online, along with its real-life adverse effects on society and public health [15], has prompted researchers to investigate what may explain people's belief in false information and conspiracy theories. In terms of demographics, being older has generally been associated with higher susceptibility to misinformation [7,9,10]. Some researchers have reported an association between self-reported minority status and belief in conspiracy theories [16,17]. Van Prooijen *et al.* [17], for example, argue that 'feelings of deprivation lead marginalized minority members to perceive the social and political system as rigged, stimulating belief in both identity-relevant and irrelevant conspiracy theories'. In addition, researchers have noted several important motivational drivers as predictors of belief in misinformation: lower trust in science and scientists [18–21], lower trust in journalists and the mainstream media [22], lower trust in government [23–25], as well as the role of political ideology, specifically conservatism [7,26–30].

In addition to these motivational factors, a growing literature has explored the role of 'cognition' in susceptibility to misinformation. Overall, a large literature finds that factors such as education [31,32], analytical thinking, numeracy skills, 'bullshit receptivity' and 'intuitive' versus 'reflective' thinking styles (often assessed via the cognitive reflection test) appear to play a consistent and key role in processing misinformation [10,14,33–39]. Following prior research [17], we included numeracy in the current study. Numeracy skills are defined as the broad ability to process basic numerical concepts and are often related to improved accuracy in judgement and decision-making across a wide range of domains [38].

Aside from these more general predictors, recent research has evaluated public belief in and susceptibility to misinformation specifically about COVID-19 [27,39–42]. A recent study by Uscinski *et al.* [41] found that beliefs in conspiracies about the virus are associated with a propensity to reject information from expert authorities, raising concerns about the potential for popular conspiracy theories to reduce people's willingness to comply with public health guidance. Accordingly, some studies have started to explore the association between belief in COVID-19 conspiracies and compliance with public health guidance [21,40,43]. Others have explored people's level of trust in politicians' and the WHO's approach to tackling the pandemic [27,41] and the role of receiving information from social media and the WHO in shaping people's beliefs about the virus [43]. Importantly, scholars have found that COVID-19 conspiracies form a 'monological belief system', where belief in one conspiracy about the virus predicts belief in others [44,45].

However, although both scientific and public interest in misinformation about COVID-19 are at a peak, no comprehensive study has yet been published that systematically examines belief in

misinformation about the virus, and which of the above-mentioned factors are most critical in shaping such beliefs. In addition, while some research is available on the relation between COVID-19 misinformation and key public health behaviours [21,40], existing studies have been criticized for using low-quality convenience samples [46] and a systematic evaluation of how false beliefs about the virus can affect vaccination-related behaviours and compliance with health guidance measures is currently lacking [47]. This is especially important considering that most research on misinformation has been confined to the USA, which limits the generalizability of prior findings [26].

Accordingly, to assess belief in misinformation and its determinants in a diverse set of countries that have experienced different death rates and government responses to the pandemic, we explore susceptibility to COVID-19 misinformation in five countries around the world: the UK, Ireland, Spain, the USA and Mexico. As detailed above, we include a broad array of known potential predictors of potential susceptibility to misinformation (both in general and about COVID-19 specifically), to conduct the first comprehensive, cross-cultural analysis of COVID-19 misinformation. Additionally, in order to examine the effects of susceptibility to misinformation about COVID-19 on self-reported behaviour, we look at two measures that are of key importance to the success of a country's approach to tackling the crisis. First, we examine the effect of higher belief in misinformation about COVID-19 on people's willingness to get vaccinated against the disease and to recommend vulnerable friends and family members to get vaccinated [40]. Second, we explore how belief in COVID-19 misinformation influences the degree of current compliance with public health guidance such as wearing a mask in public [39].[1] Our goal was not to build an exhaustive predictive model for compliance with public health guidance; rather, we sought to explore whether susceptibility to misinformation is a significant predictor of compliance with health guidance measures when controlling for other basic factors such as age, gender, education and political ideology.

## 2. Material and methods

### 2.1. Sample and procedure

In this study, we investigate susceptibility to coronavirus-related misinformation and its influence on key health-related behaviours in large national surveys in Ireland ($n = 700$), the USA ($n = 700$), Spain ($n = 700$) and Mexico ($n = 700$), conducted between mid-April and early May of 2020, and two separate surveys in the UK ($n = 1050$ and $n = 1150$). The first UK study was conducted on 14 April, and the second on 11 May 2020, in order to be able to evaluate whether the observed effects would remain constant over time. Participants were recruited as part of a series of large-scale international survey studies about COVID-19. All samples were balanced on national quotas for age and gender and obtained from Respondi, an ISO-certified panel provider of digital online data for public opinion research. Sampling continued until the quotas were filled, and the 'force response' option in Qualtrics was used to ensure complete cases for all key measures. Since the number of missing values for all measures was very low (see electronic supplementary material, table S2), we excluded missing values from the analyses. Depending on the platform and length of the survey, respondents were paid between £1.00 and £1.90 for a 20–25 minute survey.[2] Electronic supplementary material, table S2 lists the samples' demographic composition by country.

### 2.2. Measures

The measures used in this study are based on previous research about common predictors of belief in conspiracy theories and misinformation in general [10,16,17,25,36,39,48], as well as specifically about COVID-19 [23,27,39,40,43], as reviewed in the Introduction section. In terms of general predictors, participants were asked to indicate their age, gender (male, female, other or prefer not to answer), education level (1–6, 1 being 'no education above age 16' and 6 being 'doctorate'), political ideology (1–7, 1 being 'very left wing/liberal' and 7 being 'very right wing/conservative'), trust in the government, scientists and journalists (all 1–5, 1 being 'not at all' and 5 being 'very much') and 'Do you consider yourself to be part of a minority group within the country you are currently living in?' This question was phrased in this way to permit individuals who felt they were part of a minority

---

[1]The questions about vaccination are not available for the Ireland and UK surveys conducted in April 2020. For these questions, we thus only present results for Mexico, Spain, the USA and the UK surveys that were conducted in May 2020.

[2]Payments on Respondi vary between countries due to variations in compensation requirements.

group of any kind (e.g. gender, age and ideology) to answer in the affirmative. We also measured participants' numeracy level, which was calculated as the total score on three different numeracy tests: the three-item Schwartz test [49], the adaptive three-item Berlin test [50] and the answer to the following question: 'What represents the highest chance of something happening: 1 in 10, 1 in 1000, or 1 in 100' (modified from item 1 of Wright *et al.* [51]).[3] By including a broad array of numeracy questions, we sought to capture the ability of individuals to understand quantitative information in a general sense [38], which is fairly independent of intelligence and a good proxy for critical thinking [37,38].

Specifically with regard to the COVID-19 pandemic, participants were asked to what extent they comply with public health guidance ('Which of the following steps, if any, have you taken in the last month to prepare for the possibility of many cases of the coronavirus/COVID-19 in your community?'; answers included a range of options, including washing your hands, using hand sanitizer, wearing a face mask and staying at home from work). We then summed the amount of health preventative behaviours reported by each participant (out of 11), with higher scores representing higher compliance. We also asked participants whether they would get vaccinated against COVID-19 if a vaccine were to become available (yes/no); whether they would recommend their vulnerable friends to get vaccinated against COVID-19 (yes/no);[4] trust in politicians' and the World Health Organization's handling of the crisis (1–7, 1 being 'not at all' and 7 being 'very much'); their risk perception about COVID-19 (for which we used the six-item COVID-19 risk perception index developed by Dryhurst *et al.* [52])[5] and whether they have come across information about COVID-19 from the WHO and social media (both yes/no). Electronic supplementary material, table S2 shows the descriptive statistics for each of these measures by country. Electronic supplementary material, table S3 provides a detailed overview of the exact wording for all variables and items used.

To measure participants' belief in misinformation about COVID-19, we presented them with nine statements about the virus, six of which represent common examples of health-related and political misinformation (e.g. '5G networks may be making us more susceptible to the coronavirus' and 'Gargling salt water or lemon juice reduces the risk of infection from Coronavirus'), two of which were common factual statements (e.g. 'People with diabetes are at higher risk of complications from coronavirus') and one of which was not false but ambiguous (Taking ibuprofen when you are infected could make your symptoms worse).[6] The false claims were based on the World Health Organization's 'Mythbusters' page [55]. Electronic supplementary material, table S1 shows the exact wording of all items in full.

Following prior studies measuring susceptibility to misinformation [30,56], participants were asked to rate the reliability of each of these statements on a 1–7 Likert scale, from 'very unreliable' (1) to 'very reliable' (7). We averaged scores on the six misinformation items into an overall index. A reliability analysis shows good internal consistency of the six misinformation statements ($\alpha = 0.83$, $M = 2.46$, s.d. = 1.32) [44], but not for the three combined factual and ambiguous statements ($\alpha = 0.35$), so we report these separately. A multi-group confirmatory factor analysis (MG-CFA) revealed support for at least weak measurement invariance between countries.[7] Electronic supplementary material,

[3]Due to an error in the survey design, participants in Mexico and Spain were not able to answer the final question to the Schwartz numeracy test correctly. The third item of the Schwartz test was therefore left out of the calculation of the numeracy score for all countries.

[4]The two questions about vaccination intentions were not asked in the Irish and first UK survey as these were conducted first. We added these questions to the USA, Spanish, Mexican and second UK survey (conducted in May); see electronic supplementary material, table S2.

[5]Dryhurst *et al.* [52] report acceptable internal consistency for their risk perception index ($\alpha_{pooled} = 0.72$) across 10 different countries. The pooled Cronbach's $\alpha$ for the risk perception index for this study is also acceptable ($\alpha_{pooled} = 0.76$).

[6]This claim has been subject to a substantial amount of discussion in media, politics and the medical community, with medical experts usually recommending paracetamol instead of ibuprofen to treat COVID-19 symptoms [53]. On 16 April 2020, the British National Health Service issued a statement saying that 'there is no clear evidence that using ibuprofen to treat symptoms such as a high temperature can make coronavirus (COVID-19) worse' [54].

[7]To account for language differences (between the Spanish and English versions of the survey) and to ensure that any observed intergroup differences can be interpreted meaningfully, we conducted an MG-CFA for the six misinformation items, with 'country' as the grouping variable. Although we do find significant differences in the Chi-squared difference test, this test is quite sensitive to sample size. Moreover, the difference in the comparative fit index (CFI) for the weak measurement invariance model is 0.002, which is lower than the proposed cut-off point of 0.01. In addition, the difference in RMSEA is 0.021, which is between the acceptable values of 0.015 and 0.03 [57,58]; see electronic supplementary material, table S25 for a full overview. We thus find support for weak measurement invariance and, in line with Putnick & Bornstein's [59] recommendation, proceed with the analysis and interpret this as a minor limitation to our study.

table S4 contains the descriptive statistics for each item. The study was approved by the Psychology Research Ethics Committee at the Department of Psychology at the University of Cambridge.

## 2.3. Statistical analyses

To measure whether belief in misinformation about COVID-19 can be seen as a 'monological' belief system, we calculated the Pearson's correlation coefficient between the perceived reliability of six common misinformation statements about COVID-19 (see electronic supplementary material, table S1 for the full list of items). To measure susceptibility to misinformation in each country, we calculated the overall perceived reliability of six common examples of misinformation about COVID-19, pooled as a single index, for each country in the sample. We conducted a one-way analysis of variance (ANOVA) with the six-item misinformation index as the dependent variable as well as a Tukey's HSD pairwise comparison to determine whether there is a significant difference in the perceived reliability of misinformation between countries.

To investigate predictors of susceptibility to misinformation about COVID-19, we estimated an ordinary least squares (OLS) linear regression to predict susceptibility to misinformation (defined as the average of the reliability judgements of the six misinformation items) for each country. Based on the discussion in the introduction about both general and COVID-19-specific predictors of belief in misinformation, we include a number of key variables in our model. In terms of general predictors, the model controls for age, political ideology, self-perceived minority status, education, performance on numeracy tasks, trust in scientists, trust in government and trust in journalists. In terms of variables specific to the COVID-19 crisis, we include the following variables: COVID-19 risk perception, trust in politicians' and the WHO's approach to tackling the pandemic, and getting information about COVID-19 from social media and the WHO.

To investigate the effects of susceptibility to misinformation about COVID-19 on people's willingness to (i) get vaccinated against COVID-19 (yes/no), and (ii) recommend getting vaccinated to vulnerable friends or family members (yes/no), we conducted two logistic regressions with susceptibility to misinformation as the independent variable and age, gender, education level, political ideology, self-perceived minority status, numeracy score and trust in scientists as control variables.

Finally, to examine the relation between susceptibility to COVID-19 misinformation and compliance with health guidance measures, we conducted an OLS linear regression with health guidance compliance as the dependent variable, misinformation susceptibility as the independent variable, and age, gender, education level, political ideology, self-perceived minority status, numeracy score and trust in scientists as control variables. We include a variety of robustness checks—including models with robust standard errors—in the supplement.

# 3. Results

First, correlations between the perceived reliability of the six COVID-19 misinformation items range between $r = 0.288$ and $r = 0.583$ (all $p < 0.001$) and are strongest between the three conspiracy statements (about the virus being bioengineered in a Wuhan laboratory, it being part of a plot to enforce global vaccination and 5G towers exacerbating COVID-19 symptoms) ranging between $r = 0.454$ and $r = 0.583$ (see electronic supplementary material, table S20 for the full correlation matrix).

Second, figure 1 visualizes the overall perceived reliability of the six pooled misinformation statements in each country in a violin plot.

The figure shows that misinformation about the coronavirus is seen as relatively unreliable by a large majority of participants in all countries ($M_{pooled} = 2.46$, s.d. = 1.32). Strikingly, however, in all countries, the misinformation statement deemed most reliable was the claim that the coronavirus was engineered in a laboratory in Wuhan ($M_{pooled} = 3.26$, s.d. = 2.00, range $M_{UK} = 2.93$ to $M_{Spain} = 3.90$; see electronic supplementary material, table S4). Substantial segments rated this item above the midpoint of the scale (5–7), indicating that they find this conspiracy reliable, from about 22–23% in the UK and the USA, to 26% in Ireland, to about 33% and 37% in Mexico and Spain, respectively (see electronic supplementary material, table S8). In addition, we also checked how reliable participants found common factual statements about the virus (e.g. 'People with diabetes are at higher risk of complications from coronavirus'), with majorities in each country finding them reliable ($M_{diabetes} = 5.61$, s.d.$_{diabetes} = 1.52$; $M_{sanitizer} = 4.98$, s.d.$_{sanitizer} = 1.74$), and a small but persistent group rating these statements as highly unreliable.

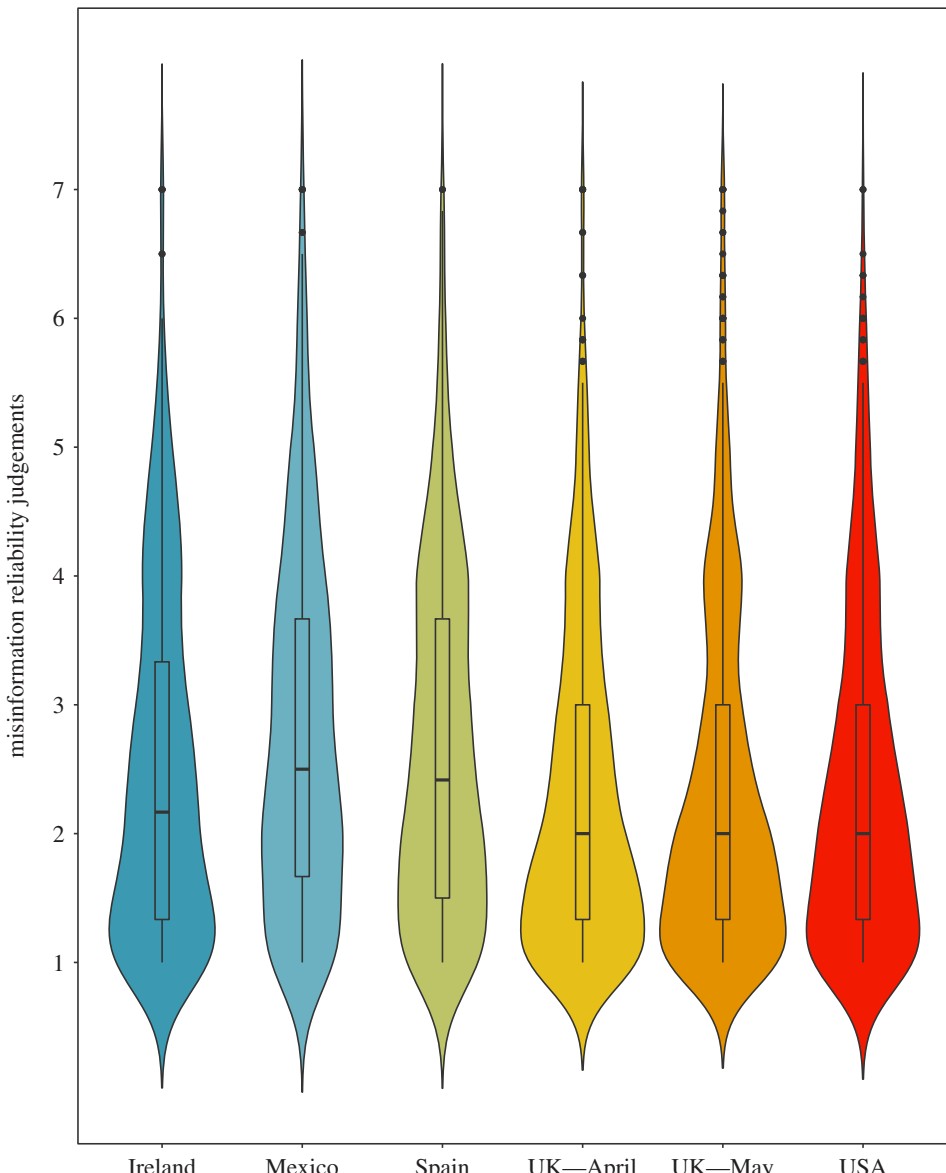

**Figure 1.** Reliability judgements of the six-item misinformation scale about COVID-19 (1–7 Likert) by country.

Using a one-way ANOVA with Tukey's HSD pairwise comparison, we find significant differences between several countries in the perceived reliability of misinformation ($F_{4,3842} = 18.87$, $p < 0.001$, $\eta^2 = 0.019$); see also electronic supplementary material, table S7. COVID-19 misinformation is perceived as the most reliable in Mexico ($M = 2.78$, s.d. $= 1.38$) and Spain ($M = 2.67$, s.d. $= 1.34$) when compared with Ireland ($M = 2.49$, s.d. $= 1.34$), the USA ($M = 2.36$, s.d. $= 1.28$) and the UK ($M_{\text{April}} = 2.31$, s.d._$_{\text{April}} = 1.25$ and $M_{\text{May}} = 2.32$, s.d._$_{\text{May}} = 1.30$; see electronic supplementary material, table S4). Furthermore, there appear to be minimal differences over time in terms of the average susceptibility to COVID-19 misinformation: as figure 1 shows, the average perceived reliability of misinformation about the virus did not significantly change between April and May in the UK ($M_{\text{diff}} = -0.012$, $t_{2,2198} = -0.225$, $p = 0.822$, $d = -0.01$); see electronic supplementary material, table S6.

## 3.1. Predictors of susceptibility to misinformation about COVID-19

Figure 2 visualizes the results of the OLS linear regression with susceptibility to misinformation as the dependent variable. Table 1 reports the regression model, pooled and by country.[8]

---

[8]For the UK, we only include the survey conducted in April in this section to maintain consistency between survey dates in each country.

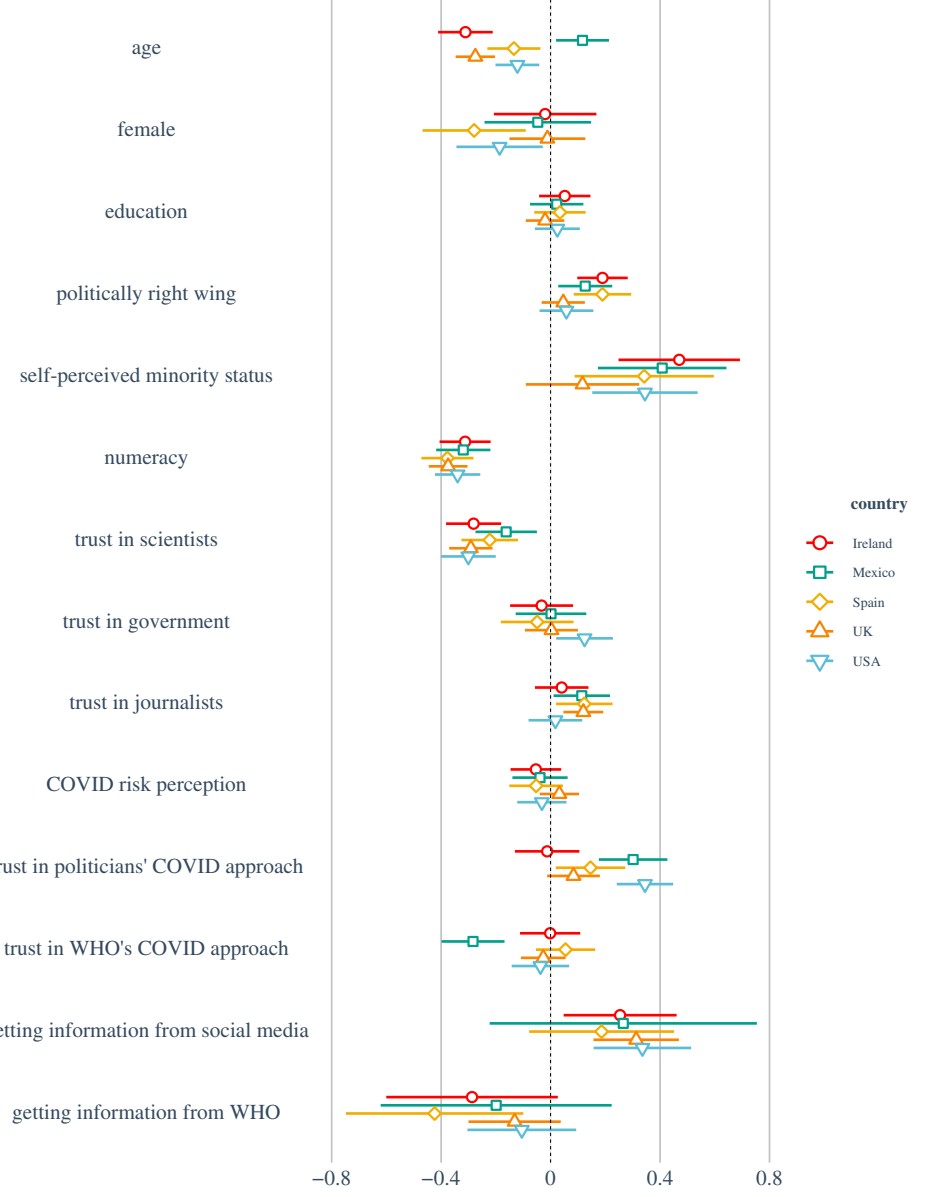

**Figure 2.** OLS multiple linear regression model for susceptibility to misinformation about COVID-19 by country. *Note*: results are mean-centred and scaled by 1 s.d. for comparability. Left of the dotted line (i.e. negative values) indicates *reduced* susceptibility to misinformation.

A number of factors stand out as significant predictors of susceptibility to misinformation across the board. Both higher performance on the numeracy tasks and higher trust in scientists are significantly and consistently associated with lower susceptibility to misinformation about COVID-19 in the pooled model and in all countries surveyed. In addition, being older is also associated with lower susceptibility to misinformation in the pooled model and in all countries, except in Mexico, where the effect is also significant but reversed.[9] Political ideology is significant in three out of five countries; identifying as more right-wing or politically conservative is associated with higher susceptibility to misinformation about COVID-19 in Ireland, Mexico and Spain, but notably not the USA and the UK. Self-identifying as a member of a minority group is also significantly associated with higher susceptibility to misinformation about the virus in all countries except the UK. Higher trust that politicians can effectively tackle the COVID-19 crisis predicts higher susceptibility to misinformation in Mexico, Spain

[9]It is worth noting that the Mexican sample was younger than the average age pooled across the other countries ($M_{Mexico} = 38.68$, s.d.$_{Mexico} = 14.57$ versus $M_{Overall} = 44.73$, s.d.$_{Overall} = 18.24$); see electronic supplementary material, table S4.

**Table 1.** Predictors of susceptibility to misinformation, pooled and by country.

| country | pooled | Ireland | Mexico | Spain | UK (April) | USA |
|---|---|---|---|---|---|---|
| N | 4733 | 643 | 644 | 664 | 1005 | 673 |
| $R^2$ | 0.23 | 0.28 | 0.21 | 0.21 | 0.24 | 0.36 |
| F | 102.11 | 17.73 | 11.75 | 12.13 | 23.76 | 27.50 |
| (intercept) | **2.30***\*\* | **2.40***\*\* | **2.64***\*\* | **2.95***\*\* | **2.17***\*\* | **2.21***\*\* |
|  | [2.21, 2.40] | [2.08, 2.73] | [2.05, 3.23] | [2.62, 3.28] | [2.00, 2.34] | [2.01, 2.42] |
| age | **−0.21***\*\* | **−0.31***\*\* | **0.12**\* | **−0.13**\*\* | **−0.28***\*\* | **−0.12**\*\* |
|  | [−0.24, −0.17] | [−0.41, −0.21] | [0.02, 0.21] | [−0.23, −0.04] | [−0.35, −0.20] | [−0.201, −0.04] |
| female | **−0.16***\*\* | −0.02 | −0.05 | **−0.28**\*\* | −0.01 | **−0.19**\* |
|  | [−0.23, −0.10] | [−0.21, 0.17] | [−0.24, 0.15] | [−0.47, −0.09] | [−0.15, 0.13] | [−0.34, −0.03] |
| education | **0.04**\* | 0.05 | 0.02 | 0.03 | −0.02 | 0.03 |
|  | [0.00, 0.07] | [−0.04, 0.15] | [−0.08, 0.12] | [−0.06, 0.13] | [−0.091, 0.050] | [−0.06, 0.11] |
| politically right wing | **0.09***\*\* | **0.19***\*\* | **0.13**\* | **0.19***\*\* | 0.05 | 0.06 |
|  | [0.05, 0.12] | [0.10, 0.28] | [0.03, 0.23] | [0.09, 0.30] | [−0.03, 0.13] | [−0.04, 0.16] |
| self-perceived minority status | **0.38***\*\* | **0.47***\*\* | **0.41***\*\* | **0.34**\*\* | 0.12 | **0.35***\*\* |
|  | [0.29, 0.47] | [0.25, 0.69] | [0.17, 0.64] | [0.1, 0.60] | [−0.09, 0.32] | [0.15, 0.54] |
| numeracy | **−0.40***\*\* | **−0.31***\*\* | **−0.32***\*\* | **−0.38***\*\* | **−0.37***\*\* | **−0.34***\*\* |
|  | [−0.43, −0.36] | [−0.41, −0.22] | [−0.42, −0.22] | [−0.47, −0.28] | [−0.45, −0.30] | [−0.42, −0.26] |
| trust in scientists | **−0.24***\*\* | **−0.28***\*\* | **−0.16**\*\* | **−0.22***\*\* | **−0.29***\*\* | **−0.30***\*\* |
|  | [−0.27, −0.20] | [−0.38, −0.18] | [−0.28, −0.05] | [−0.33, −0.12] | [−0.37, −0.21] | [−0.40, −0.20] |
| trust in government | −0.01 | −0.03 | 0.00 | −0.05 | 0.00 | **0.12**\* |
|  | [−0.05, 0.04] | [−0.15, 0.08] | [−0.13, 0.13] | [−0.18, 0.08] | [−0.09, 0.10] | [0.02, 0.23] |
| trust in journalists | **0.11***\*\* | 0.04 | **0.11**\* | **0.12**\* | **0.12**\*\* | 0.02 |
|  | [0.07, 0.15] | [−0.06, 0.14] | [0.01, 0.22] | [0.02, 0.23] | [0.05, 0.19] | [−0.08, 0.12] |
| COVID risk perception | −0.02 | −0.05 | −0.04 | −0.05 | 0.03 | −0.03 |
|  | [−0.06, 0.01] | [−0.15, 0.04] | [−0.14, 0.06] | [−0.15, 0.05] | [−0.04, 0.10] | [−0.12, 0.06] |
| trust in politicians' COVID approach | **0.14***\*\* | −0.01 | **0.30***\*\* | **0.15**\* | 0.08 | **0.35***\*\* |
|  | [0.10, 0.19] | [−0.13, 0.11] | [0.18, 0.43] | [0.02, 0.27] | [−0.01, 0.18] | [0.24, 0.45] |
| trust in WHO's COVID approach | **−0.08***\*\* | −0.00 | **−0.28***\*\* | 0.06 | −0.03 | −0.04 |
|  | [−0.12, −0.04] | [−0.11, 0.11] | [−0.40, −0.17] | [−0.05, 0.16] | [−0.11, 0.06] | [−0.14, 0.07] |
| getting information from social media | **0.35***\*\* | **0.25**\* | 0.27 | 0.19 | **0.31***\*\* | **0.34***\*\* |
|  | [0.27, 0.43] | [0.05, 0.46] | [−0.22, 0.75] | [−0.08, 0.45] | [0.16, 0.47] | [0.16, 0.51] |
| getting information from WHO | **−0.14**\*\* | −0.29 | −0.20 | **−0.42**\* | −0.13 | −0.11 |
|  | [−0.23, −0.04] | [−0.60, 0.03] | [−0.62, 0.22] | [−0.75, −0.10] | [−0.30, 0.04] | [−0.30, 0.09] |

*Note*: \*$p < 0.05$, \*\*$p < 0.01$, \*\*\*$p < 0.001$. Beta values are standardized and 95% confidence intervals are provided in parentheses. Significant predictors are marked in bold.[10]

and the USA. Finally, being exposed to information about the virus on social media is significantly associated with higher susceptibility to misinformation in Ireland, the UK and the USA. Electronic supplementary material, tables S13–S19 show the correlation matrices per country.

## 3.2. Susceptibility to misinformation about COVID-19 and behavioural outcomes

To investigate the relation between misinformation and self-reported behaviour, we first examined whether susceptibility to COVID-19 misinformation influences people's willingness to get vaccinated against the virus and recommend getting vaccinated to others. Figure 3 shows the results for the logistic regression for whether people would get vaccinated against COVID-19 themselves (the results for whether they would recommend vaccination to friends and family are highly similar, see

[10]The pooled sample contains both the April and May UK surveys. To evaluate the robustness of the model, we checked for linearity violations and whether the residuals are normally distributed (see electronic supplementary material, figure S1), and ran a robust standard error regression for the pooled model as well as by country, finding no meaningful differences between the robust model and the model reported here. The Durbin–Watson test (1.93, $p = 0.004$) confirmed that the errors were relatively uncorrelated (see electronic supplementary material, table S12).

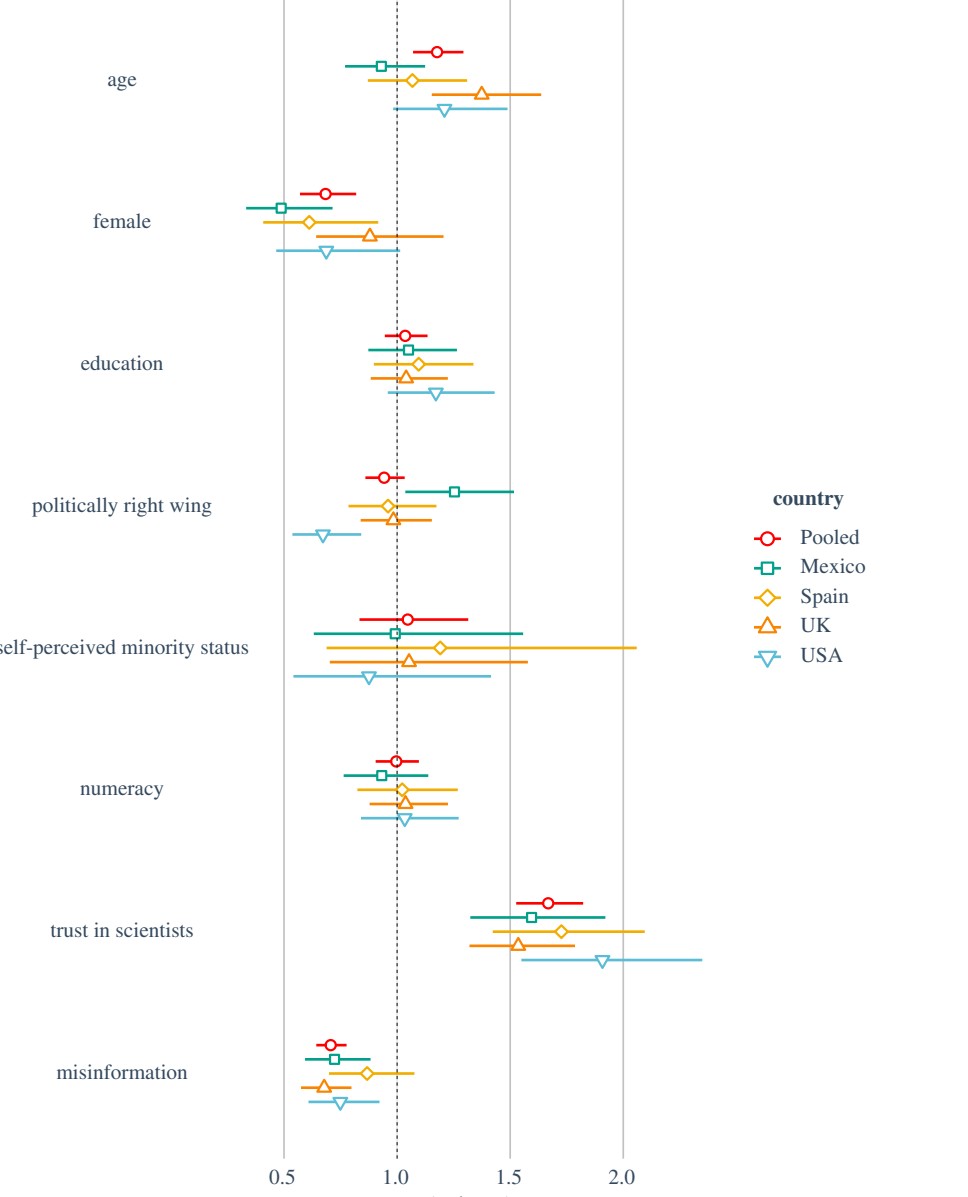

**Figure 3.** Logistic regression model for 'Would you get vaccinated against COVID-19 (y/n)' by country. *Note*: coefficients are exponentiated and represent odds ratios. Left of the dotted line (i.e. values less than 1) indicates *reduced* likelihood to get vaccinated.

electronic supplementary material, figure S2). Electronic supplementary material, tables S9 and S10 list the individual regression models by country.

Figure 3 shows that, when controlling for all other factors, a one-unit increase in susceptibility to misinformation (which was measured on a 1–7 Likert scale) is associated with a 23% (OR = 0.77, 95% CI [0.72, 0.83]) and 28% (OR = 0.72, 95% CI [0.67, 0.78]) decrease in the likelihood to get vaccinated and to recommend vaccination to vulnerable friends and family, respectively (see electronic supplementary material, table S10). This effect is consistent in all countries (with odds ratios ranging from 0.68 to 0.80) except Spain. Conversely, being older, male and especially having higher trust in scientists are all associated with an increased likelihood to get vaccinated against COVID-19, as well as to recommend others to get vaccinated. Some of these effects are substantial; for example, a one-unit increase in trust in scientists is associated with a 73% (OR = 1.73, 95% CI [1.57–1.91]) increase in the odds of getting vaccinated and a 79% (OR = 1.79, 95% CI [1.61–1.98]) increase in the odds of recommending vaccination to others. Education level, political ideology, self-identified minority status and performance on the numeracy task were not significant predictors of vaccination intentions (all $p > 0.21$).

**Table 2.** Predictors of compliance with health guidance, pooled and by country.

| country | pooled | Ireland | Mexico | Spain | UK | USA |
|---|---|---|---|---|---|---|
| N | 4745 | 644 | 645 | 665 | 1007 | 673 |
| $R^2$ | 0.08 | 0.07 | 0.11 | 0.06 | 0.07 | 0.12 |
| F | 53.11 | 7.21 | 10.02 | 6.52 | 8.73 | 12.66 |
| (intercept) | 6.96*** | 6.87*** | 8.23*** | 7.42*** | 6.45*** | 7.23*** |
| | [6.85, 7.10] | [6.59, 7.15] | [7.95, 8.51] | [7.12, 7.72] | [6.22, 6.68] | [6.92, 7.55] |
| age | **0.20***  | **0.35***  | **0.53***  | 0.09 | **0.23** | 0.20 |
| | [0.13, 0.28] | [0.15, 0.55] | [0.35, 0.71] | [−0.12, 0.30] | [0.07, 0.40] | [−0.01, 0.40] |
| female | **0.86***  | **1.09***  | **0.70***  | **0.62** | **0.948***  | **1.00***  |
| | [0.71, 1.00] | [0.72, 1.46] | [0.33, 1.06] | [0.21, 1.03] | [0.64, 1.254] | [0.60, 1.41] |
| education | **0.25***  | 0.01 | 0.05 | 0.07 | **0.19*** | **0.26*** |
| | [0.17, 0.32] | [−0.17, 0.20] | [−0.14, 0.23] | [−0.14, 0.27] | [0.03, 0.35] | [0.05, 0.47] |
| politically right wing | 0.03 | −0.11 | 0.05 | **0.40***  | **0.20*** | −0.14 |
| | [−0.04, 0.11] | [−0.29, 0.07] | [−0.13, 0.24] | [0.20, 0.61] | [0.04, 0.35] | [−0.36, 0.08] |
| self-perceived minority status | **0.35***  | **0.56*** | 0.20 | 0.40 | 0.23 | 0.05 |
| | [0.15, 0.54] | [0.11, 1.02] | [−0.24, 0.65] | [−0.16, 0.96] | [−0.23, 0.70] | [−0.45, 0.54] |
| numeracy | **0.10*** | **0.26** | 0.12 | 0.03 | 0.16 | 0.12 |
| | [0.02, 0.18] | [0.07, 0.45] | [−0.07, 0.31] | [−0.19, 0.25] | [−0.01, 0.33] | [−0.10, 0.34] |
| trust in scientists | **0.44***  | 0.07 | **0.31** | **0.37***  | **0.23** | **0.52***  |
| | [0.36, 0.51] | [−0.12, 0.26] | [0.12, 0.50] | [0.16, 0.58] | [0.07, 0.40] | [0.30, 0.75] |
| misinformation susceptibility | **−0.17***  | −0.10 | **−0.42***  | **−0.35** | −0.01 | **−0.28*** |
| | [−0.26, −0.09] | [−0.31, 0.10] | [−0.61, −0.22] | [−0.58, −0.13] | [−0.18, 0.17] | [−0.51, −0.05] |

Note: *$p < 0.05$, **$p < 0.01$, ***$p < 0.001$. Beta values are standardized and 95% confidence intervals are provided in parentheses. Significant predictors are marked in bold.[10]

Next, we investigated whether susceptibility to COVID-19 misinformation predicts compliance with common health guidelines. Table 2 shows the results of the OLS linear regression with compliance with health guidance as the dependent variable.

Higher susceptibility to misinformation is the only variable in the model that predicts *lower* compliance with public health guidance, particularly in Mexico and Spain, but also in the USA. This effect is not significant in Ireland and the UK. Also of note is that women are significantly more likely than men to comply with a greater number of public health guidance measures in all countries surveyed.

# 4. Discussion

Overall, we find that the majority of people in the countries we surveyed do not report finding misinformation about COVID-19 credible, consistent with research on the spread of fake news in other contexts [7–10]. However, we do find some cross-cultural differences, with the Mexican and Spanish samples rating COVID-19 misinformation as most reliable, before Ireland, the UK and the USA. This result follows a similar pattern as participants' performance on the numeracy task across these five countries, which may explain the variation (see electronic supplementary material, table S5). Nonetheless, notable segments in all countries find misinformation reliable. For example, between 7.43% (for the statement about hot air, in the USA) and 37.57% (for the statement about the virus being bioengineered in a laboratory in Wuhan, in Spain; see electronic supplementary material, table S8) of the sample rated misinformation about the virus as at least somewhat reliable, which is consistent with other recent estimates from the USA and the UK [6,13,23,41,60]. The strong correlations observed between these statements also support the idea that belief in misinformation about COVID-19 can be seen as a 'monological belief system' where belief in one conspiracy correlates with belief in others [31,44,45].

Moreover, although at most about 16% of people in our sample find the 5G conspiracy reliable (i.e. rate it above the midpoint, particularly in Spain and Mexico), the consequences of acting on those beliefs

can be dire for society, as belief in the 5G conspiracy has been linked to violent intentions [6]. In fact, scholars have referred to misinformation about COVID-19 as a 'meta-risk' interfering with people's initial risk perception of the virus [61], which in itself is linked to the adoption of preventative health behaviours [52].

## 4.1. Predictors of susceptibility to misinformation about COVID-19

We demonstrate that certain factors significantly influence people's susceptibility to misinformation about COVID-19. For example, being exposed to information about the virus on social media is associated with higher susceptibility to misinformation,[11] which is consistent with recent research showing that people are more likely to encounter fake news (including about COVID-19) via social media [9,43]. One explanation for this finding is that misinformation via social media may be amplified through social consensus [62]. Further research is therefore needed to investigate the proliferation of COVID-19-related misinformation in specific social media echo chambers [43]. In addition, consistent with much previous research [7,10,22,23,27–30,41,63], we find that political conservatism is associated with a slightly higher susceptibility to misinformation. However, although the direction is consistent, we did not find this association in the USA and in the UK, which is surprising given that right-leaning outlets were more likely to spread misinformation about COVID-19 in the early stages of the pandemic [64]. This could partially be explained by the fact that some of the variables in the model may have mediated the influence of political ideology; for example, 'trust in government' and 'trust in politicians' COVID-approach' are both political variables, and both are significant predictors in the USA (but not in the UK).

With respect to age, we find that being older is significantly associated with lower susceptibility to misinformation in all countries except in Mexico. This is not consistent with prior research [7,9,10], which typically finds the opposite pattern at least in the context of elections. For example, Guess *et al.* [10] found that being older than 65 was the largest predictor of sharing political fake news online. Importantly, it should be noted that in these studies, the context is different (politics versus health, e.g. older people are more vulnerable to COVID-19), and the variable of interest is often sharing of fake news, which we did not measure here. Nonetheless, it may be possible that while older individuals are less susceptible, they still share more fake news, for motivations other than accuracy (e.g. political gain and social consensus).

A number of other findings are more difficult to interpret. For example, self-identifying as a member of a minority predicts susceptibility to misinformation about the virus in all countries surveyed, except the UK (contrary to findings by Guess *et al.* [10]). It is important to note here that our question was phrased so as to include *any* self-identified minority group, not only ethnicity. For example, (strong) partisans could also self-identify as a minority, but models without self-perceived minority status did not substantially alter the effect of ideology on susceptibility to misinformation.[12] Recent research suggests that conspiracy beliefs are more common among 'those who are more marginalized, reflected by lower levels of psychological well-being, education, and income' [16,17,23].[13] It is also possible that misinformation proliferates more widely in some networks than others, perhaps as a result of coordinated disinformation campaigns [65]. Some members of communities that are exposed to a larger than average amount of misinformation may, in turn, begin to perceive it as more reliable due to the 'illusory truth effect' [14]. At any rate, we caution against oversimplifying these findings, especially as we were unable to control for several other potentially important factors such as income or religiosity [34].

On the other hand, several factors appear to significantly *decrease* susceptibility to COVID-19 misinformation. First, higher trust in scientists is associated with lower belief in misinformation in all countries. This highlights not just the critical role that scientists play in combating the virus [21], but also the importance of communicating scientific research to the public: if the communication of

---

[11]This effect is not significant in Mexico and Spain, but we note the low number of people who indicate *not* receiving news about COVID-19 on social media, especially in Mexico ($n = 30$), as well as the relatively large confidence intervals around these estimates.

[12]To check for potential multi-collinearity issues between minority status and other variables in the main analyses (i.e. susceptibility to misinformation, vaccine hesitancy and compliance with health guidance measures), we ran the same analysis with the self-perceived minority status variable excluded. This gives approximately the same results, with little variation. For example, being politically right-wing and self-perceived minority status do not appear to suffer from multi-collinearity. See electronic supplementary material, tables S21–S23 for a full overview of each analysis without the minority variable included.

[13]It is worth noting that self-identified minorities in our sample did not have a lower level of education than non-minorities.

scientific information is perceived as an attempt to be open and transparent [66], it might be trusted more and thereby reduce reliance on misinformation. This is especially important in light of the finding that higher trust in scientists is also associated with a higher willingness to get vaccinated against COVID-19 or to recommend vaccination to vulnerable friends and family, as well as compliance with health guidance measures. Considering that trust is an important predictor of attention to recommendations by scientific experts [67], demonstrating trustworthiness when giving scientific expertise may thus have multiple beneficial outcomes.

Second, we note an interesting dichotomy in terms of the sources by which people acquire information about COVID-19 [43]: although getting information about the virus from social media is associated with *higher* susceptibility to misinformation, getting information from the WHO is associated with *lower* susceptibility. This finding alludes to the importance of source selection: the social media information landscape allows for a wide range of opinions to be expressed, which may not be subject to fact-checks or gatekeeping before being posted [68]. Choosing not to acquire information about COVID-19 from social media may thus reduce the amount of unofficial information that people receive, which in turn could reduce belief in misinformation.

Finally, the most consistent predictor of decreased susceptibility to misinformation about COVID-19 is performance on the numeracy tasks. It is worth noting that the construct of numeracy does not merely measure mathematical ability but captures the ability of individuals to understand and use quantitative information more broadly [38] and is associated with a propensity to apply a more 'system 2'-reliant mode of critical thinking [37]. Although some literature finds that higher numeracy facilitates rather than protects against motivated cognition [37], our findings are consistent with a large literature which finds that reflective and analytical thinking are consistently associated with reduced susceptibility to misinformation [33,34,69]. These findings are promising for potential interventions and suggest that developing critical thinking skills—such as learning how to identify fake news—may be an effective strategy to combat misinformation about COVID-19 [39,70–72].

## 4.2. COVID-19 misinformation and its influence on public health behaviours

Although previous research has debated the societal consequences of fake news [62], we clearly show that susceptibility to misinformation can be a significant factor in influencing people's behaviour during the COVID-19 outbreak in three important ways: it may make people less likely to report willingness to get vaccinated against COVID-19,[14] it may make them less likely to recommend vaccination to vulnerable people in their social circle, and it may decrease people's willingness to comply with public health guidance measures [39]. These findings are consistent with other emerging research in the context of COVID-19 which has linked specific conspiracy theories to lower willingness to adopt public health behaviours in the USA, France and the UK [39–41]. They also highlight the critical importance of limiting the spread of misinformation about the virus. For example, increased vaccine hesitancy can reduce vaccination rates and compromise herd immunity [75]. Previous research suggests that whereas *post hoc* corrections may backfire [76], pre-emptive refutations of conspiracy theories through a process known as 'cognitive inoculation' can be effective at reducing belief in misinformation [29,30,77,78].

Of course, our study is not without limitations. First, although we report robust associations across different cultural contexts in large national samples, we cannot infer causality. Although we have sought to include variables that are known to be important factors in shaping people's belief in misinformation in our models, we note that these results are exploratory and correlational. Having said this, it is unlikely that, for example, rating misinformation as more reliable *causes* lower numeracy; it is more likely that lower numeracy, representing, perhaps, a low propensity to engage in effortful 'system 2' type critical thinking, contributes to higher susceptibility to misinformation. Nonetheless, we cannot rule out alternative explanations. Similarly, although we have modelled misinformation as a predictor of vaccine hesitancy consistent with prior work in this area [39,41,42], we note the likely possibility that causality can run both ways: being vaccine hesitant may, in turn,

---

[14]We note that we cannot disentangle the causal direction of effects in this study. Both options are plausible, i.e., belief in COVID-19 misinformation could reduce willingness to get vaccinated, and prior vaccine hesitancy could increase belief in misinformation [73,74]. A supplementary linear regression with misinformation as the dependent variable and with the question 'would you get vaccinated against COVID-19' as an independent variable (along with age, education, gender, political ideology, minority status, numeracy skills, COVID-19 risk perception and trust in scientists) shows that being willing to get vaccinated against the virus is a significant predictor of lower susceptibility to misinformation in three out of four countries (Spain being the exception). See electronic supplementary material, table S24 for a full overview.

also lead people to become more susceptible to misinformation (see electronic supplementary material, table S24). We encourage future research to investigate potential feedback loops between information and behaviour, and disentangle the direction of causality using non-recursive, longitudinal or experimental data.

Second, although our samples were balanced on national quotas and are therefore of higher quality than common convenience samples, they were not true probability samples of the target population in each country, and between-country differences in susceptibility to misinformation should therefore be interpreted with caution. Third, we note that the 'compliance with health guidance measures' variable suffers from some limitations: participants were not asked if they were voluntarily performing certain preventative behaviours (for example, if they were not staying home from work because their profession does not allow them to); and our measure does not take into account variability in certain behaviours (such as how often people wash their hands) due to its binary nature (participants could indicate either complying with a certain measure or not). However, we note that our results are similar to recent work on the relation between misinformation and health guidance compliance which does not suffer from the same limitations [47], indicating that our measure captures compliance with some degree of accuracy. Fourth, as we report in footnote 7 and electronic supplementary material, table S25, the six-item misinformation index is not entirely measurement non-invariant (although we find decent evidence for at least weak measurement invariance). We therefore caution the reader when interpreting differences between countries reported in this study. Notwithstanding these limitations, we offer the first large-scale comparative study of predictors of susceptibility to misinformation about COVID-19 and its link to preventative health behaviours.

# 5. Conclusion

We present the results of an international study, integrating previous research about predictors of belief in misinformation (both in general and specifically about COVID-19), and, in turn, how susceptibility to misinformation about the virus affects key self-reported health behaviours. In summary, while belief in misinformation about COVID-19 is not held by a majority of people in any country that we examined, specific misinformation claims are consistently deemed reliable by a substantial segment of the public and pose a potential risk to public health. Crucially, we demonstrate a clear link, replicated internationally, between susceptibility to misinformation and vaccine hesitancy and a reduced likelihood of complying with public health guidance. We highlight the key role that scientists play as disseminators of factual and reliable information, as well as the potential importance of fostering numeracy and critical thinking skills as a way to reduce susceptibility to misinformation. Further research should explore how digital media and risk literacy interventions may impact how (mis)information is received, processed and shared, and how they can be leveraged to improve resilience against misinformation on a societal level.

Ethics. This study was approved by the Psychology Research Ethics Committee at the Department of Psychology at the University of Cambridge.

Data accessibility. All data are available in the manuscript or the supplementary materials, and have been uploaded to the OSF: https://osf.io/jnu74/.

Authors' contributions. J.R. and S.v.d.L. conceptualized the study. A.L.J.F., S.v.d.L., J.R., S.D., C.R.S., G.R., J.K. and A.M.v.d.B. designed the instrument. A.L.J.F., G.R. and J.K. collected and processed the data. J.R. conducted all of the analyses with input from S.v.d.L., C.R.S., S.D., A.L.J.F. and G.R.; J.R. and S.v.d.L. drafted the manuscript with input and revisions from C.R.S., A.L.J.F., S.D., G.R., J.K. and A.M.v.d.B.

Competing interests. We declare we have no competing interests.

Funding. We received no funding for this study.

Acknowledgements. The Spanish translation of the survey was conducted by María del Carmen Climént-Palmer. We thank all participants for their time and input. The surveys in the UK, the USA, Mexico and Spain were funded by the Winton Centre for Risk & Evidence Communication, which is supported by a donation from the David & Claudia Harding Foundation. The survey in Ireland was funded by Science Foundation Ireland.

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
