## [Reviewer comments · Royal Society Open Science]

Review History

RSOS-201199.R0 (Original submission)

Review form: Reviewer 1

Is the manuscript scientifically sound in its present form?

Yes

Are the interpretations and conclusions justified by the results?

Yes

Is the language acceptable?

Yes

Do you have any ethical concerns with this paper?

No

Have you any concerns about statistical analyses in this paper?

No

Recommendation?

Accept with minor revision (please list in comments)

Comments to the Author(s)

First of all, I would like to commend the authors for tackling an important topic (i.e. misinformation about COVID-19) in a very thorough way by ensuring several nationally representative samples from different cultural backgrounds and including several predictors as well as outcomes of misinformation. The manuscript is very well written and easy to understand. I also appreciate the fact that the authors have agreed to make the data available via OSF. However, I believe that the paper would benefit from some (mostly minor) changes prior to its publication, such as providing additional information in some parts, an even clearer structure in some sections and possibly some additional analyses. My comments/suggestions are arranged by sections.

Abstract:

- Page 1, lines 23-24 ("In five nationally representative studies"): in hindsight, I think that the emphasis on different samples instead of studies is more appropriate here, since the paper does not have a classical multiple study structure, but rather employs different samples to investigate the same research questions

- Page 1, lines 25-26 ("we examine predictors of belief in the most common statements ..."): while the abstract is generally very well-written, I believe that the aim of the study is presented too narrowly here. In reality, the paper not only examines predictors of belief in misinformation about COVID-19, but also the prevalence of misinformation and the role of susceptibility to misinformation as a predictor of COVID-19-related behavior

Introduction:

- General: the introduction is generally thorough, yet concentrated and concise. However, the authors should expand on the (potential) role of numeracy skills. Currently, the introduction provides a very convincing argument for many of the variables included, whereas numeracy skills are not even mentioned. As a reader, I have no idea at this stage that this variable is being measured, and I do not understand why it would make sense to include it, which is not ideal since many of the important and interesting results revolve around this variable. Something similar could be said about minority status. Please justify their inclusion in the introduction (even if the study was exploratory, there has to be some justification for why these variables were considered as potential correlates)

-Page 2, line 6 ("emerged in December 2019"): I suggest an alternative wording because it is possible that the first human infection occurred in November, if not earlier. It is of course true that the first reported case occurred in December 2019

-Page 2, lines 11-12: I suggest updating the numbers when revising the manuscript, since the numbers change so rapidly

Materials and methods:

- General: I have some questions/comments regarding the compliance scale: 1) Were the participants asked whether they were freely able to perform preventive behaviors? In other words, did you include any control questions regarding the external barriers to participants' behavior (e.g. the "staying at home" item could only be answered with "yes" among individuals with occupations that can be performed from home and the "no" answer does not necessarily mean that a person is non-compliant). Otherwise, I suggest mentioning this in the limitations sections. 2) Another possible limitation of this scale is its dichotomous nature. For example, I would imagine that there is limited variability in some of the responses (e.g., washing hands), and one could argue whether the "number of behaviors performed" really captures compliance, since the extent to which such behaviors are performed is also very important. Please clarify this a bit more and perhaps mention it in the limitations section. 3) Please explain the origins of the items (were they derived from any official guidelines?) and how the answers were scored, as this is not completely unambiguous (e.g. buying extra supplies or food could be an indicator of compliance as you visit the shops less often, but it could also be an indicator of non-compliance - hoarding supplies)

- Page 5, lines 16-18: as the dates/timeframes are provided for the UK samples, it would also be useful to explain when the data were collected in the remaining countries (even if the results show only limited changes over time). Provide this information here instead of in results
- Page 5, lines 24-25: since online panels were used for data collection, it would be useful to find out more about any quality metrics (if provided by Respondi) and/or any precautions that were taken to ensure the quality of data (e.g. attention checks, speeding traps, ... ; see, for example, Kees, Berry, Burton, & Sheehan, 2017)
Kees, J., Berry, C., Burton, S., & Sheehan, K. (2017). An analysis of data quality: Professional panels, student subject pools, and Amazon's Mechanical Turk. *Journal of Advertising*, 46(1), 141-155.
- Page 5, line 27: what did the payment depend on? Why did it differ among individuals? Explain it in the text or in a footnote, as it is unclear at the moment
- General: it is a little unclear to me why some questions were left out in some of the samples. This should be explained in more detail (I did not see it mentioned in the text at all, but some data are missing in table S2 as well as the database)
- General: more information about internal consistencies of the scales is needed, particularly the risk perception scale. Please provide Cronbach's alpha or arguments which explain why alpha was not calculated
- General: the authors should consider including the "Statistical analyses" section to the Method and hence removing some analyses information from the Results section

Results:

- General: the results are interesting and well presented (the visuals are nice and intuitive). I did find it somewhat odd that the results 1) contain a lot of "model building" (i.e. explaining why the variables were included, see, for example, page 11, lines 1-20; this could be moved to earlier sections) and 2) are already compared to previous literature (this could be moved to later sections; the same applies to sentences referring to the contributions of the study). In my opinion, this would improve the flow and make the results section more coherent
- General: some parts of the results revolve around comparing different countries. Since both the English version of the questionnaires and the Spanish version of the questionnaires were administered, invariance testing is needed to ensure that the differences can be interpreted meaningfully. See for example the paper by Milfont and Fischer (2010)
Milfont, T. L., & Fischer, R. (2010). Testing measurement invariance across groups: Applications in cross-cultural research. *International Journal of Psychological Research*, 3(1), 111-130.
- Page 13: it was a bit unclear to me that the pooled column contains both UK samples, but the UK column contains only one of the two UK samples. Please add a line to clarify this
- Table S2: since many of the included tables have a pooled column, it would also make sense to add it in this table and thus describe the characteristics of the whole sample
- Please provide more information on how you have dealt with missing values somewhere in the manuscript

Discussion and conclusion:

- No comments; I found both of these sections very clear and concise

Review form: Reviewer 2

Is the manuscript scientifically sound in its present form?

Yes

Are the interpretations and conclusions justified by the results?

Yes

Is the language acceptable?

Yes

Do you have any ethical concerns with this paper?

No

Have you any concerns about statistical analyses in this paper?

Yes

Recommendation?

Accept with minor revision (please list in comments)

Comments to the Author(s)

This study presents a comprehensive cross-national test of prominent hypotheses about susceptibility to misinformation in the context of Covid-19. I see it as a significant contribution to the burgeoning literature on this important topic. Two items I would like to see addressed, however:

- I have concerns about the self-perceived minority status measure. Aside from some of the qualifications mentioned in the text, it seems plausible that partisans could view themselves as minorities, confounding any estimates. An example of this is Republicans in the U.S., a substantial portion of whom view discrimination against whites as a significant problem in society. At the very least, I would like to see versions of the primary analyses without this measure included (and in particular, I'm interested to see whether this changes the coefficient on right-wing ideology due to possible multicollinearity).

- I'm concerned about the models with vaccine intention and public health compliance behavior as the DVs because they include misinformation susceptibility as a predictor. As the preceding analysis suggests, susceptibility is partially endogenous to other factors. It would be helpful to redo these analyses as analogues to Table 1/Figure 2 in which these are swapped in as DVs (i.e., keep susceptibility out entirely). There's an implied causal ordering in which misinfo -> behavior but I note that it's not tested here and there are plausible reasons to expect the reverse.

Decision letter (RSOS-201199.R0)

Dear Dr Roozenbeek

The Editors assigned to your paper RSOS-201199 "Susceptibility to misinformation about COVID-19 around the world" have now received comments from reviewers and would like you to revise the paper in accordance with the reviewer comments and any comments from the Editors. Please note this decision does not guarantee eventual acceptance.

We do not generally allow multiple rounds of revision so we urge you to make every effort to fully address all of the comments at this stage. If deemed necessary by the Editors, your

manuscript will be sent back to one or more of the original reviewers for assessment. If the original reviewers are not available, we may invite new reviewers.

Please submit your revised manuscript and required files (see below) no later than 21 days from today's (ie 27-Aug-2020) date. Note: the ScholarOne system will 'lock' if submission of the revision is attempted 21 or more days after the deadline. If you do not think you will be able to meet this deadline please contact the editorial office immediately.

on behalf of Dr Christina Demski (Associate Editor) and Essi Viding (Subject Editor)
openscience@royalsociety.org

Associate Editor Comments to Author (Dr Christina Demski):

Comments to the Author:

Both reviewers agree the manuscript is in good shape but requires some revisions before it can be accepted. Please address all comments clearly and comprehensively.

Reviewer comments to Author:

Reviewer: 1

Comments to the Author(s)

First of all, I would like to commend the authors for tackling an important topic (i.e. misinformation about COVID-19) in a very thorough way by ensuring several nationally representative samples from different cultural backgrounds and including several predictors as well as outcomes of misinformation. The manuscript is very well written and easy to understand. I also appreciate the fact that the authors have agreed to make the data available via OSF. However, I believe that the paper would benefit from some (mostly minor) changes prior to its publication, such as providing additional information in some parts, an even clearer structure in some sections and possibly some additional analyses. My comments/suggestions are arranged by sections.

Abstract:

- Page 1, lines 23-24 ("In five nationally representative studies"): in hindsight, I think that the emphasis on different samples instead of studies is more appropriate here, since the paper does not have a classical multiple study structure, but rather employs different samples to investigate the same research questions

- Page 1, lines 25-26 ("we examine predictors of belief in the most common statements ..."): while the abstract is generally very well-written, I believe that the aim of the study is presented too narrowly here. In reality, the paper not only examines predictors of belief in misinformation about COVID-19, but also the prevalence of misinformation and the role of susceptibility to misinformation as a predictor of COVID-19-related behavior

Introduction:

- General: the introduction is generally thorough, yet concentrated and concise. However, the authors should expand on the (potential) role of numeracy skills. Currently, the introduction provides a very convincing argument for many of the variables included, whereas numeracy skills are not even mentioned. As a reader, I have no idea at this stage that this variable is being measured, and I do not understand why it would make sense to include it, which is not ideal since many of the important and interesting results revolve around this variable. Something similar could be said about minority status. Please justify their inclusion in the introduction (even if the study was exploratory, there has to be some justification for why these variables were considered as potential correlates)

-Page 2, line 6 ("emerged in December 2019"): I suggest an alternative wording because it is possible that the first human infection occurred in November, if not earlier. It is of course true that the first reported case occurred in December 2019

-Page 2, lines 11-12: I suggest updating the numbers when revising the manuscript, since the numbers change so rapidly

Materials and methods:

- General: I have some questions/comments regarding the compliance scale: 1) Were the participants asked whether they were freely able to perform preventive behaviors? In other words, did you include any control questions regarding the external barriers to participants' behavior (e.g. the "staying at home" item could only be answered with "yes" among individuals with occupations that can be performed from home and the "no" answer does not necessarily mean that a person is non-compliant). Otherwise, I suggest mentioning this in the limitations sections. 2) Another possible limitation of this scale is its dichotomous nature. For example, I would imagine that there is limited variability in some of the responses (e.g., washing hands), and one could argue whether the "number of behaviors performed" really captures compliance, since the extent to which such behaviors are performed is also very important. Please clarify this a bit more and perhaps mention it in the limitations section. 3) Please explain the origins of the items (were they derived from any official guidelines?) and how the answers were scored, as this is not completely unambiguous (e.g. buying extra supplies or food could be an indicator of compliance as you visit the shops less often, but it could also be an indicator of non-compliance - hoarding supplies)

- Page 5, lines 16-18: as the dates/timeframes are provided for the UK samples, it would also be useful to explain when the data were collected in the remaining countries (even if the results show only limited changes over time). Provide this information here instead of in results

- Page 5, lines 24-25: since online panels were used for data collection, it would be useful to find out more about any quality metrics (if provided by Respondi) and/or any precautions that were taken to ensure the quality of data (e.g. attention checks, speeding traps, ... ; see, for example, Kees, Berry, Burton, & Sheehan, 2017)

Kees, J., Berry, C., Burton, S., & Sheehan, K. (2017). An analysis of data quality: Professional panels, student subject pools, and Amazon's Mechanical Turk. *Journal of Advertising*, 46(1), 141-155.

- Page 5, line 27: what did the payment depend on? Why did it differ among individuals? Explain it in the text or in a footnote, as it is unclear at the moment

- General: it is a little unclear to me why some questions were left out in some of the samples. This should be explained in more detail (I did not see it mentioned in the text at all, but some data are missing in table S2 as well as the database)

- General: more information about internal consistencies of the scales is needed, particularly the risk perception scale. Please provide Cronbach's alpha or arguments which explain why alpha was not calculated

- General: the authors should consider including the "Statistical analyses" section to the Method and hence removing some analyses information from the Results section

Results:

- General: the results are interesting and well presented (the visuals are nice and intuitive). I did find it somewhat odd that the results 1) contain a lot of "model building" (i.e. explaining why the variables were included, see, for example, page 11, lines 1-20; this could be moved to earlier sections) and 2) are already compared to previous literature (this could be moved to later sections; the same applies to sentences referring to the contributions of the study). In my opinion, this would improve the flow and make the results section more coherent

- General: some parts of the results revolve around comparing different countries. Since both the English version of the questionnaires and the Spanish version of the questionnaires were administered, invariance testing is needed to ensure that the differences can be interpreted meaningfully. See for example the paper by Milfont and Fischer (2010)

Milfont, T. L., & Fischer, R. (2010). Testing measurement invariance across groups: Applications in cross-cultural research. *International Journal of Psychological Research*, 3(1), 111-130.

- Page 13: it was a bit unclear to me that the pooled column contains both UK samples, but the UK column contains only one of the two UK samples. Please add a line to clarify this

-Table S2: since many of the included tables have a pooled column, it would also make sense to add it in this table and thus describe the characteristics of the whole sample

-Please provide more information on how you have dealt with missing values somewhere in the manuscript

Discussion and conclusion:

- No comments; I found both of these sections very clear and concise

Reviewer: 2

Comments to the Author(s)

This study presents a comprehensive cross-national test of prominent hypotheses about susceptibility to misinformation in the context of Covid-19. I see it as a significant contribution to the burgeoning literature on this important topic. Two items I would like to see addressed, however:

- I have concerns about the self-perceived minority status measure. Aside from some of the qualifications mentioned in the text, it seems plausible that partisans could view themselves as minorities, confounding any estimates. An example of this is Republicans in the U.S., a substantial portion of whom view discrimination against whites as a significant problem in society. At the very least, I would like to see versions of the primary analyses without this measure included (and in particular, I'm interested to see whether this changes the coefficient on right-wing ideology due to possible multicollinearity).

- I'm concerned about the models with vaccine intention and public health compliance behavior as the DVs because they include misinformation susceptibility as a predictor. As the preceding analysis suggests, susceptibility is partially endogenous to other factors. It would be helpful to redo these analyses as analogues to Table 1/Figure 2 in which these are swapped in as DVs (i.e., keep susceptibility out entirely). There's an implied causal ordering in which misinfo -> behavior but I note that it's not tested here and there are plausible reasons to expect the reverse.

===PREPARING YOUR MANUSCRIPT===

a 'clean' version of the new manuscript that incorporates the changes made, but does not highlight them. This version will be used for typesetting if your manuscript is accepted. Please ensure that any equations included in the paper are editable text and not embedded images.

===PREPARING YOUR REVISION IN SCHOLARONE===

<https://royalsociety.org/journals/authors/author-guidelines/#supplementary-material> to include a suitable title and informative caption. An example of appropriate titling and captioning may be found at [https://figshare.com/articles/Table_S2_from_Is_there_a_trade-off_between_peak_performance_and_performance_breadth_across_temperatures_for_aerobic_sc ope_in_teleost_fishes_/3843624](https://figshare.com/articles/Table_S2_from_Is_there_a_trade-off_between_peak_performance_and_performance_breadth_across_temperatures_for_aerobic_scope_in_teleost_fishes_/3843624).

Author's Response to Decision Letter for (RSOS-201199.R0)

See Appendix A.

RSOS-201199.R1 (Revision)

Review form: Reviewer 1

Is the manuscript scientifically sound in its present form?

Yes

Are the interpretations and conclusions justified by the results?

Yes

Is the language acceptable?

Yes

Do you have any ethical concerns with this paper?

No

Have you any concerns about statistical analyses in this paper?

No

Recommendation?

Accept as is

Comments to the Author(s)

I would like to thank the authors for addressing my concerns. I believe that the manuscript is now scientifically sound, very clearly written and it represents an important contribution to the existing literature.

Review form: Reviewer 2**Is the manuscript scientifically sound in its present form?**

Yes

Are the interpretations and conclusions justified by the results?

Yes

Is the language acceptable?

Yes

Do you have any ethical concerns with this paper?

No

Have you any concerns about statistical analyses in this paper?

No

Recommendation?

Accept as is

Comments to the Author(s)

I am satisfied with these changes. Thanks to the authors for their revisions, which I think have improved the manuscript.

Decision letter (RSOS-201199.R1)

Dear Dr Roozenbeek,

It is a pleasure to accept your manuscript entitled "Susceptibility to misinformation about COVID-19 around the world" in its current form for publication in Royal Society Open Science. The comments of the reviewer(s) who reviewed your manuscript are included at the foot of this letter.

COVID-19 rapid publication process:

We are taking steps to expedite the publication of research relevant to the pandemic. If you wish, you can opt to have your paper published as soon as it is ready, rather than waiting for it to be published the scheduled Wednesday.

This means your paper will not be included in the weekly media round-up which the Society sends to journalists ahead of publication. However, it will still appear in the COVID-19

Publishing Collection which journalists will be directed to each week (<https://royalsocietypublishing.org/topic/special-collections/novel-coronavirus-outbreak>).

If you wish to have your paper considered for immediate publication, or to discuss further, please notify openscience_proofs@royalsociety.org and press@royalsociety.org when you respond to this email.

on behalf of Dr Christina Demski (Associate Editor) and Essi Viding (Subject Editor)
openscience@royalsociety.org

Associate Editor Comments to Author (Dr Christina Demski):

Thank you for comprehensively addressing the reviewer comments. Both reviewers have now indicated that they are satisfied with the revisions and that the manuscript is improved as a result.

Reviewer comments to Author:
Reviewer: 1
Comments to the Author(s)

I would like to thank the authors for addressing my concerns. I believe that the manuscript is now scientifically sound, very clearly written and it represents an important contribution to the existing literature.

Reviewer: 2
Comments to the Author(s)

I am satisfied with these changes. Thanks to the authors for their revisions, which I think have improved the manuscript.

Appendix A

Associate Editor Comments to Author (Dr Christina Demski):

Comments to the Author:

Both reviewers agree the manuscript is in good shape but requires some revisions before it can be accepted. Please address all comments clearly and comprehensively.

Dear Dr. Demski,

First of all, thank you for coordinating this timely and very useful review. We have done our best to address all comments thoroughly, and we believe that the reviewers' comments and suggestions have significantly improved our manuscript. We wish to thank them for their efforts, and hope that the paper is now found ready for publication. For convenience, all comments to the reviewers are marked in yellow below. In addition, we provide an updated manuscript, with all changes marked in yellow as well.

Warm regards,

Dr. Jon Roozenbeek

Reviewer comments to Author:

Reviewer: 1

Comments to the Author(s)

First of all, I would like to commend the authors for tackling an important topic (i.e. misinformation about COVID-19) in a very thorough way by ensuring several nationally representative samples from different cultural backgrounds and including several predictors as well as outcomes of misinformation. The manuscript is very well written and easy to understand. I also appreciate the fact that the authors have agreed to make the data available via OSF. However, I believe that the paper would benefit from some (mostly minor) changes prior to its publication, such as providing additional information in some parts, an even clearer structure in some sections and possibly some additional analyses. My comments/suggestions are arranged by sections.

We very much appreciate that the reviewer recognises the importance and urgency of the topic and the value of high-quality cross-cultural and open data on this subject. We sincerely thank the reviewer for their helpful and constructive review. We have gone through all of the comments carefully and made several significant changes to our manuscript, which we think has significantly improved it in terms of clarity, style, and rigour. Please find our comments below, point by point. All changes in the manuscript are marked in yellow for convenience.

Abstract:

- Page 1, lines 23-24 ("In five nationally representative studies"): in hindsight, I think that the emphasis on different samples instead of studies is more appropriate here, since the paper does not have a classical multiple study structure, but rather employs different samples to investigate the same research questions

We completely agree and are thankful to the reviewer for pointing this out. We have changed the wording to "5 national samples" in the abstract and throughout the paper where appropriate.

- Page 1, lines 25-26 ("we examine predictors of belief in the most common statements ..."): while

the abstract is generally very well-written, I believe that the aim of the study is presented too narrowly here. In reality, the paper not only examines predictors of belief in misinformation about COVID-19, but also the prevalence of misinformation and the role of susceptibility to misinformation as a predictor of COVID-19-related behavior

Thank you; this is a great point. We have now amended the abstract, and included the following sentence: 'We also investigate the prevalence of belief in such misinformation across different countries, and the role of belief in COVID-19 misinformation in predicting relevant public health behaviours.'

Introduction:

- General: the introduction is generally thorough, yet concentrated and concise. However, the authors should expand on the (potential) role of numeracy skills. Currently, the introduction provides a very convincing argument for many of the variables included, whereas numeracy skills are not even mentioned. As a reader, I have no idea at this stage that this variable is being measured, and I do not understand why it would make sense to include it, which is not ideal since many of the important and interesting results revolve around this variable. Something similar could be said about minority status. Please justify their inclusion in the introduction (even if the study was exploratory, there has to be some justification for why these variables were considered as potential correlates)

Thank you, we completely understand this point and this was an oversight on our part. There is a large literature on the role of "cognition" or "analytical thinking" (vs ideology) as predictors of belief in misinformation, but we did not make the link between numeracy and these other measures sufficiently clear. Previous research has used indicators such as education and the Cognitive Reflection Test (CRT) as measures of 'analytical' thinking. We wanted to add to this literature by using a conceptually related but different measure of critical thinking: numeracy. Numeracy is often used in the science polarization literature (e.g., see Kahan et al., 2012) and we included the numeracy measure as a proxy for critical/analytical thinking (e.g. see Peters et al. (2006), "Numeracy and Decision Making", *Psychological Science* 17(5), 407-413). We have now made the role of numeracy in susceptibility to misinformation much clearer on the front end of the manuscript as you are correct that it is one of the most important determinants in our model and should therefore have been described more clearly. Accordingly, we have now added a justification for including the numeracy questions on page 7.

-Page 2, line 6 ("emerged in December 2019"): I suggest an alternative wording because it is possible that the first human infection occurred in November, if not earlier. It is of course true that the first reported case occurred in December 2019

Thank you for this clarification; we fully agree that this is possible, but the WHO reports that the first reported case occurred in early December of 2019 (as the reviewer mentions). We have reworded the first sentence to match wording provided by the WHO: 'The first human infection with the SARS CoV-2 novel coronavirus (COVID-19) was **reported** in December of 2019 in Wuhan, China.'

-Page 2, lines 11-12: I suggest updating the numbers when revising the manuscript, since the numbers change so rapidly

Yes, thank you! We have updated the numbers to the most recent figures.

Materials and methods:

- General: I have some questions/comments regarding the compliance scale: 1) Were the participants asked whether they were freely able to perform preventive behaviors? In other words, did you include any control questions regarding the external barriers to participants' behavior (e.g. the "staying at home" item could only be answered with "yes" among individuals with occupations that can be performed from home and the "no" answer does not necessarily mean that a person is non-compliant). Otherwise, I suggest mentioning this in the limitations sections.

2) Another possible limitation of this scale is its dichotomous nature. For example, I would imagine that there is limited variability in some of the responses (e.g., washing hands), and one could argue whether the "number of behaviors performed" really captures compliance, since the extent to which such behaviors are performed is also very important. Please clarify this a bit more and perhaps mention it in the limitations section.

3) Please explain the origins of the items (were they derived from any official guidelines?) and how the answers were scored, as this is not completely unambiguous (e.g. buying extra supplies or food could be an indicator of compliance as you visit the shops less often, but it could also be an indicator of non-compliance – hoarding supplies)

We are thankful to the reviewer for pointing out these issues. These are all good points and indeed some limitations of our measure. We did unfortunately not ask about external barriers to performing the behaviours (good point) and the items themselves were broadly based on the WHO's advice. We also agree that the dichotomous nature of the scale offers limited variability in the responses. In terms of measurement, our scale measure simply counted the number of health-protective behaviours the respondent indicated to engage in with higher numbers broadly indicating better compliance (see page 7).

In response, we have now done three things: 1) we have clarified the measure (p. 7), 2) we now acknowledge the limitations of our scale (p. 24); and 3) we also find that our results are highly similar to other recent studies that have looked at the relation between belief in misinformation and health guidance compliance that do take into account the variability within certain behaviours and thus generally do not suffer from the same limitations as our measure (e.g., Allington et al., 2020: <https://doi.org/10.1017/S003329172000224X>). Thus, while we acknowledge that the measure we used is imperfect, we nonetheless find it encouraging that our results are in agreement with other recent work. To address this, we have added a section to our discussion of the limitations of our study (p. 23/24) that acknowledges these limitations.

- Page 5, lines 16-18: as the dates/timeframes are provided for the UK samples, it would also be useful to explain when the data were collected in the remaining countries (even if the results show only limited changes over time). Provide this information here instead of in results

Thank you! We have added a clarification about the survey dates to the 'sample and procedure' section (p.5).

- Page 5, lines 24-25: since online panels were used for data collection, it would be useful to find out more about any quality metrics (if provided by Respondi) and/or any precautions that were taken to ensure the quality of data (e.g. attention checks, speeding traps, ... ; see, for example, Kees, Berry, Burton, & Sheehan, 2017)

Kees, J., Berry, C., Burton, S., & Sheehan, K. (2017). An analysis of data quality: Professional panels, student subject pools, and Amazon's Mechanical Turk. *Journal of Advertising*, 46(1), 141-155.

We only know that the Respondi panel is ISO-certified (20252:2019), which (unlike other platforms such as Prolific or Mturk) is an industry specific standard for quality market, opinion, and social research and includes audits concerning the collection and reporting of the data. We have clarified this certification in the manuscript (page 6). Because we did not pre-register any exclusion criteria, selectively excluding participants post-hoc based on attention checks and speeding is generally not recommended (Ejelov & Luke 2020, JESP) due to the fairly arbitrary nature of these cut-offs. In short, although we did not implement such measures here, general inspection of the data did not indicate unusual responding or speeding patterns (e.g., "77777").

- Page 5, line 27: what did the payment depend on? Why did it differ among individuals? Explain it in the text or in a footnote, as it is unclear at the moment

Thank you for pointing this out; Respondi's payment system is somewhat complicated, as their payments vary per country (based on average wages and compensation requirements). We have added an explanation of this on page 6.

- General: it is a little unclear to me why some questions were left out in some of the samples. This should be explained in more detail (I did not see it mentioned in the text at all, but some data are missing in table S2 as well as the database)

Thank you, good point. The Irish and first UK study were conducted first. After the data was collected, we realised that it would be very useful and interesting to also include several questions about vaccine hesitancy so we could examine the link with misinformation, which we then included in the subsequent US, Spanish, Mexican, and second UK surveys. We have added a footnote on page 7 clarifying this decision.

- General: more information about internal consistencies of the scales is needed, particularly the risk perception scale. Please provide Cronbach's alpha or arguments which explain why alpha was not calculated

This is a good point; Dryhurst et al. (2020), whose risk perception scale we used, report a pooled Cronbach's alpha of 0.72 (ranging between 0.60 and 0.82) across 10 different countries for their risk perception index, indicating acceptable internal consistency. We have added the internal consistency of the risk perception index for our sample as well ($\alpha = 0.76$). We have added a footnote on page 7 to clarify this: 'Dryhurst et al. [49] report acceptable internal consistency for their risk perception index ($\alpha_{\text{pooled}} = 0.72$) across 10 different countries. The pooled Cronbach's α for the risk perception index for this study is 0.76.'

- General: the authors should consider including the "Statistical analyses" section to the Method and hence removing some analyses information from the Results section

We fully agree with this. We have now added a 'statistical analyses' section (page 9), and removed explanations of the model and statistical methods from the 'results' section. In our view, this has greatly improved the readability of the paper; thank you.

Results:

- General: the results are interesting and well presented (the visuals are nice and intuitive). I did find

it somewhat odd that the results 1) contain a lot of "model building" (i.e. explaining why the variables were included, see, for example, page 11, lines 1-20; this could be moved to earlier sections) and 2) are already compared to previous literature (this could be moved to later sections; the same applies to sentences referring to the contributions of the study). In my opinion, this would improve the flow and make the results section more coherent

Thank you, and we agree. We have moved the explanation for the primary regression model forward (it is now explained in the 'statistical analyses' section; see p. 9). In addition, we have expanded our discussion about the purpose of our study on page 4/5; this was initially placed later in the paper (in the 'results' section), but we agree with the reviewer that this is better placed in the introduction. We have now also reserved comparisons of results to previous literature for the discussion section.

- General: some parts of the results revolve around comparing different countries. Since both the English version of the questionnaires and the Spanish version of the questionnaires were administered, invariance testing is needed to ensure that the differences can be interpreted meaningfully. See for example the paper by Milfont and Fischer (2010)

Milfont, T. L., & Fischer, R. (2010). Testing measurement invariance across groups: Applications in cross-cultural research. *International Journal of Psychological Research*, 3(1), 111-130.

Thank you; this was indeed an oversight on our part. We have added a multi-group confirmatory factor analysis to the supplement (Supplementary Table S25) to evaluate measurement invariance, which shows that while the chi squared differences test is significant (which is sensitive to differences in sample size), we find acceptable values for the difference in Comparative Fit Index (CFI) and difference in the Root Mean Square Error of Approximation (RMSEA), indicating support for at least weak measurement invariance. Generally, we follow Putnick & Bornstein's (2016) (<https://doi.org/10.1016/j.dr.2016.06.004>) recommendation that:

"Minor deviations from invariance could be stated as a limitation of the study, and group differences could be interpreted accordingly. The concern is that potentially important comparative research will never see the light of print if full invariance cannot be achieved. Without solid research on the real-life implications of noninvariance, we see rejecting all noninvariant models as premature. Instead, we encourage researchers to test invariance, report their results and interpret any deviations from invariance in the context of the construct, test group differences if it makes sense to do so, and report any limitations of the tests."

Accordingly, we have added a footnote on page 8 to explain the above. In addition, we have added information about measurement invariance to our limitations section (page 24) to (despite some support for invariance) be sufficiently cautious about any country-level comparisons.

- Page 13: it was a bit unclear to me that the pooled column contains both UK samples, but the UK column contains only one of the two UK samples. Please add a line to clarify this

Much appreciated; we have clarified this on page 14.

-Table S2: since many of the included tables have a pooled column, it would also make sense to add it in this table and thus describe the characteristics of the whole sample

Thank you for this suggestion. We have added a pooled column to Supplementary Table S2.

-Please provide more information on how you have dealt with missing values somewhere in the manuscript

Thank you; we used “force-response” for most key items and so missing data was very low (mostly < % 0.5) and were therefore simply excluded from the analysis. We have added more information about this on page 6 (the “sample and procedure” section).

One exception is that both questions about vaccination are shown in table S2 as having a large number of missing values; but this is due to the fact that these questions were not asked in the Irish and first UK surveys. We have clarified this in footnote 14 on page 36.

Discussion and conclusion:

- No comments; I found both of these sections very clear and concise

Thank you!

We hope that the reviewer now finds the manuscript acceptable for publication.

Reviewer: 2

Comments to the Author(s)

This study presents a comprehensive cross-national test of prominent hypotheses about susceptibility to misinformation in the context of Covid-19. I see it as a significant contribution to the burgeoning literature on this important topic. Two items I would like to see addressed, however:

We thank the reviewer for their positive assessment and for reading our manuscript in such detail, we very much appreciate the constructive comments. As the reviewer mentions, we hoped to offer a more comprehensive cross-cultural contribution to the literature. Based on the reviewers’ recommendation, we have now implemented several additional analyses, which we believe has helped us improve the manuscript significantly. Please find our comments and changes below. For your convenience, all changes in the manuscript are marked in yellow.

- I have concerns about the self-perceived minority status measure. Aside from some of the qualifications mentioned in the text, it seems plausible that partisans could view themselves as minorities, confounding any estimates. An example of this is Republicans in the U.S., a substantial portion of whom view discrimination against whites as a significant problem in society. At the very least, I would like to see versions of the primary analyses without this measure included (and in particular, I'm interested to see whether this changes the coefficient on right-wing ideology due to possible multicollinearity).

We thank the reviewer for pointing this out. We had significantly debated the nature of this measure ourselves as technically it is *self-identified* minority status so indeed it is possible that anyone can include themselves in this measure. However, at a first glance, a cross-tab between self-reported minority status and ideology on the US data does not seem to indicate that Conservatives/Republicans were more likely to identify as a minority in the United States; please find the cross-tab below for convenience. In fact, if anything, liberals seem to be more likely to self-report as having minority status. Nonetheless, as per the reviewers’ request we now have re-run the 3 main analyses (with susceptibility to misinformation, vaccine hesitancy and compliance with health

guidance measures as the DVs) with the self-perceived minority status variable excluded. This gives approximately the same results; political ideology, for example, remains a similarly significant predictor across all 3 regression analyses when minority status is not included in the model. We have added a footnote linking to this analysis on page 23. The supplementary analyses can be found in Supplementary Tables S21-23. We agree with the reviewer that this was an interesting hypothesis but it seems political conservatism and self-perceived minority status are not collinear. The bivariate correlation is also negative, indicating that minority status is more associated with political liberalism/identifying as left-wing in the US and fairly unrelated to ideology in the pooled data ($r_{\text{pooled}} = -0.059$, $r_{\text{US}} = -0.207$, both $ps < 0.001$).

Country	Political ideology	Self-identified minority status			Total
		No	Yes	Prefer not to answer	
United States	Very left wing/liberal	44	24	1	69
	Left wing/liberal	65	30	0	95
	Centre left/slightly liberal	50	20	4	74
	Middle of the road	161	56	11	228
	Centre right/slightly conservative	86	9	3	98
	Right wing/conservative	69	7	1	77
	Very right wing/conservative	49	6	0	55
	Total	524	152	20	696
Pooled	Very left wing/liberal	198	80	8	286
	Left wing/liberal	593	148	18	759
	Centre left/slightly liberal	609	142	23	774
	Middle of the road	1436	329	97	1862
	Centre right/slightly conservative	580	88	17	685
	Right wing/conservative	390	60	13	463
	Very right wing/conservative	104	39	0	143
	Total	3910	886	176	4972

- I'm concerned about the models with vaccine intention and public health compliance behavior as the DVs because they include misinformation susceptibility as a predictor. As the preceding analysis suggests, susceptibility is partially endogenous to other factors. It would be helpful to redo these analyses as analogues to Table 1/Figure 2 in which these are swapped in as DVs (i.e., keep susceptibility out entirely). There's an implied causal ordering in which misinfo -> behavior but I note that it's not tested here and there are plausible reasons to expect the reverse.

We are thankful to the reviewer for drawing our attention to this. Before building the regression model (with misinformation susceptibility as the DV), we spent quite some time deciding whether to include compliance with public health guidance as an IV. In the end, we decided against it, as we did not find it entirely plausible that health compliance would be a causal factor to misinformation belief (meaning: it does not seem quite likely that complying less with health guidance would induce higher belief in misinformation). We thought it more plausible that the relationship would work the other way around (i.e., belief in misinformation might make a person less willing to comply with health guidance measures). This is consistent with theory and a range of prior studies which have included misinformation as the independent variable (e.g., Bertin et al., 2020; Freeman et al., 2020; Imhoff & Lamberty, 2020; Stanley et al., 2020). However, we fully agree with the reviewer that for vaccine

hesitancy the causal relationship could plausibly go both ways, as prior vaccine hesitancy may also lead to more belief in misinformation. The reason why we did not include willingness to get vaccinated against COVID-19 as an IV in the linear regression is because this question was not included in the Irish and first UK surveys, and we wanted to ensure that we could compare all predictors across all 5 countries on the full sample.

Nonetheless, we highly appreciate your drawing attention to the reversed causality issue (that is of course inherent in nearly all cross-sectional data). We have therefore added a supplementary analysis with misinformation as the DV and vaccine hesitancy included for the Mexican, Spanish, US and second UK samples. You were right to point out that the reverse is also true; we find that while belief in misinformation predicts reduced willingness to get vaccinated (as reported in our study), the reverse holds true as well in all countries except Spain (in the sense that indicating being willing to get vaccinated predicts lower belief in misinformation). We have added discussion in the limitations section that now fully acknowledges the causality issue for the reader and the difficulties with inferring causality from cross-sectional data. Although ultimately the relationship between misinformation and hesitancy is likely dynamic and bi-directional and will require more complicated longitudinal or experimental data to disentangle, our specific interest here is in establishing whether misinformation is also a significant predictor of vaccine hesitancy and public health compliance (it may not be as some scholars claim that fake news is inconsequential). Yet, the relationship appears significant and robust across countries (but we now acknowledge the reverse is also likely true). The full regression table can be found in Supplementary Table S24.

We hope that we have sufficiently addressed the excellent concerns raised and that the reviewer can now recommend the manuscript for publication.